# L–asparaginase activity in some endophytic fungi: Glutaminase–free and low urease co–activities

Zahra Zaeimian[1], Khalil-Berdi Fotouhifar[1]*, Mohsen Farzaneh[2]

**1** Department of Plant Protection, Faculty of Agriculture, College of Agriculture and Natural Resources, University of Tehran, Karaj, Iran, **2** Department of Agriculture, Medicinal Plants and Drugs Research Institute, Shahid Beheshti University, Tehran, Iran

* fotowhi@ut.ac.ir

## Abstract

In this study, L–asparaginase production in several endophytic fungi was evaluated along with their L–glutaminase and urease co-activities. The effect of L–asparagine and different culture media on L-asparaginase production were also evaluated. Among the 62 investigated isolates, 49 isolates exhibited L–asparaginase activity, and the maximum zone index (6.58) was observed in *Cladosporium perangustum* EL1. Evaluation of L–glutaminase and urease co-activities in L–asparaginase-positive isolates screened 19 isolates with no L–glutaminase activity and four isolates with minimum urease production. L–asparaginase activity was quantified in 12 selected isolates using the Nesslerization method. *Cladosporium cladosporioides* Kr5−2 exhibited the maximum L–asparaginase activity (10.78 U mL$^{-1}$). *Alternaria brassicae* C showed high L–asparaginase activity (7.07 U mL$^{-1}$) free of L–glutaminase, and low urease co-activity (1.97 U mL$^{-1}$). Assessment of the effect of L–asparagine on L–asparaginase activity showed that the enzyme is inducible and substrate-regulated. Evaluation of ten different culture media showed that all isolates were able to produce L–asparaginase on Mineral salts agar and Citrate agar culture media. Also, Cerelose ammonium nitrate agar, Kuehner basal culture medium, and Piefer, Humphrey, and Acree culture medium inhibited L–asparaginase production in the majority of the isolates. This is the first report of L–asparaginase production by endophytic fungi isolated from *Taxus baccata*, *Pistacia vera*, *Prunus avium*, *Prunus cerasus*, and *Punica granatum*, as well as the investigation of their L–glutaminase and urease co-activities. Among the evaluated culture media, Mineral salts agar and Citrate agar culture media are suggested here as alternate for MCD. Moreover, *Alternaria brassicae* C is recommended as a promising isolate for future commercial L–asparaginase production.

**Data availability statement:** All relevant data are within the manuscript and its Supporting Information files.

**Funding:** The author(s) received no specific funding for this work.

**Competing interests:** The authors have declared that no competing interests exist.

## Introduction

Acute lymphoblastic leukemia (ALL) is a type of cancer that occurs predominantly in children [1]. Although it is also seen in adults, it comprises approximately 25% of cancers diagnosed in children, especially between the ages of 2 and 4 years [1–4]. Tumor cells in ALL depend on circulating L–asparagine in the bloodstream, which is produced by healthy cells and can be obtained from the diet [5]. The importance of exocellular L–asparagine for the growth and proliferation of leukemic leukocytes is due to the fact that some tumor cells are negative-asparagine synthetase, the enzyme that converts aspartate to asparagine, and some of positive-asparagine synthetase do not express an adequate amount of this enzyme to produce L–asparagine endogenously [3,6–8]. Hence, the use of L–asparaginase (ASNase) in the clinical protocol, particularly in pediatric ALL treatment, as an auxiliary chemotherapeutic agent accompanied by other drugs such as vincristine and dexamethasone, has increased the overall 5–year survival rate in children to 90% [3,7,9]. This therapeutic enzyme catalyzes the hydrolysis of L–asparagine into aspartate and ammonia and decreases the concentration of L–asparagine in the blood serum [6,10,11]. Since L-asparagine is involved in DNA, RNA, and protein synthesis, its unavailability for tumor cells leads to cellular starvation and eventually apoptosis [3,12]. The application of L–asparaginase (ASNase) does not affect healthy cells due to their expression of asparagine synthetase and ability to produce L-asparagine [7,13]. Currently, three types of L–asparaginase (ASNase) are used in the treatment of ALL. These include Elspar® and Oncaspar® (the native and PEGylated forms of L–asparaginase derived from *Escherichia coli*) and Erwinaze® (L–asparaginase obtained from *Erwinia chrysanthemi*) [14,15]. The bacterial L–asparaginase, which is currently used in medicine, has several disadvantages [3]. Hypersensitivity and silent inactivation are the main issues that prevent patients from receiving complete treatment [7]. Furthermore, the L–glutaminase (GLNase) and urease co-activities of L–asparaginase leads to significant side effects. Hepatotoxicity, hyperglycemia, pancreatitis, neurological seizures, thrombosis, immunosuppression, and leukopenia are among the common effects reported in several studies [3,6,10,16–18]. In addition to these side effects, the L–glutaminase (GLNase) and urease co-activities can result in hyperammonemia by hydrolyzing glutamine and urea in the blood, potentially leading to neurotoxicity [4,6,19]. Due to the complications associated with bacterial L–asparaginase, researchers are seeking alternative sources for L–asparaginase production [17,20]. L–asparaginase is found in various biological sources, including animals, plants, algae, and microorganisms [21]. Although humans possess L–asparaginase enzyme, its $K_m$ value is not low enough to allow effective asparagine depletion [7,17]. Among L–asparaginase producers, filamentous fungi are considered as reliable and promising sources [22,23]. The commercial L–asparaginases – Acrylaway® (from *Aspergills oryzae* strain NZYM–SP and strain NZYM-OA, produced by Novozymes, Denmark) and PreventAse™ (from *Aspergillus niger* strain DS 53180, produced by DSM Food Specialties, Denmark) – have been approved by the Food and Drug Administration (FDA) and are used in the food industry to reduce acrylamide formation [24–27]. Endophytic fungi

are microorganisms that inhabit apparently healthy plant tissues for at least part of their life cycle without causing any disease symptoms [28]. Due to their eukaryotic nature, endophytic fungi produce L–asparaginase with post–translational modifications that are more compatible with the human body and may reduce immunological problems [4,7,12,29]. Their extracellular enzyme secretion enables easier and more cost–effective extraction and purification of L–asparaginase compared to bacterial counterparts [13]. It has been established that endophytic fungi can produce L–asparaginase without L–glutaminase (GLNase) and urease co–activities [4]. It has been demonstrated that L–asparaginase of eukaryotic origin may be more suitable for pharmaceutical applications [30]. Therefore, in the current study, several endophytic fungi from five different plant species (*Taxus baccata*, *Pistacia vera*, *Prunus avium*, *Prunus cerasus*, and *Punica granatum*) were screened for L–asparaginase production, as well as for the absence of L–glutaminase and urease co-activities, for the first time. Moreover, the effect of different culture media on L–asparaginase production of the investigated fungal isolates was also evaluated, and two novel culture media for screening of L–asparaginase production were proposed. Furthermore, the effect of L–asparagine as a substrate on the relevant enzyme activity was also evaluated.

## Materials and methods

### Endophytic fungi

In the present study, endophytic fungal isolates were obtained from the culture collection of the Department of Plant Protection, College of Agriculture and Natural Resources, University of Tehran. These isolates were obtained from *Taxus baccata* L., *Pistacia vera* L., *Prunus avium* L., *Prunus cerasus* L., and *Punica granatum* L. from 2013 to 2017 in Iran. The morphological and molecular characterization of their representative isolates had been performed previously [31–34]. The isolates used in the present study were originated from the previous studies. Their candidates had been thoroughly characterized based on the morphological features, as well as their ITS1–5.8S–ITS2 rDNA region had also been sequenced and the resulte sequences have been deposited in the GeneBank (NCBI) database [https://www.ncbi.nlm.nih.gov]. The characteristics of the candidates of the studied isolates, including their names, host sources, and accession numbers, are presented in the S1 Table. The endophytic fungal isolates were re-cultured on potato dextrose agar (PDA) and re-purified using the hyphal tip or single spore technique. Moreover, their identity was confirmed by morphological re–examination.

### Qualitative assay of L-asparaginase production

Screening of endophytic fungi for L–asparaginase production was performed according to the method described by Gulati *et al.* [35], using the Modified Czapek Dox (MCD) agar culture medium (g L$^{-1}$): Glucose 2, L–asparagine 10, KH$_2$PO$_4$ 1.52, KCl 0.52, MgSO$_4$·7H$_2$O 0.52, FeSO$_4$·7H$_2$O 0.01, ZnSO$_4$·7H$_2$O 0.01, CuNO$_3$·3H$_2$O 0.01 and agar 18. A 0.4% (w/v) stock solution of bromothymol blue (BTB), used as a pH indicator, was prepared in 70% ethanol, and 0.01% of this solution was added to the culture medium. The pH of the culture medium was adjusted to 5.5 using 4M NaOH [36]. The mycelial discs were prepared from the margin of seven-day-old endophytic fungal colonies and inoculated onto MCD culture medium. The plates were incubated in the dark at $26\pm2°C$. After five days, the diameter of the blue color zone formed around the colony was measured, indicating the potential of the fungal isolates to produce L–asparaginase. The zone index was calculated as the ratio of the diameter of the color zone to that of the fungal colony. The experiment was conducted in triplicate for each endophytic fungal isolate. MCD culture medium without L–asparagine was used as the negative control [37–39].

### Evaluation of L–glutaminase and urease co–activities

Endophytic fungal isolates capable of producing L–asparaginase were subjected to L–glutaminase and urease assays. The procedures followed the above–mentioned method, except that L–glutamine (for L–glutaminase assay) and filter–sterilized urea solution (for urease assay) were used as substrates. All assays were performed in triplicate. [36,38].

 

## Quantitative assay of L-asparaginase activity

The L-asparaginase activity of selected endophytic fungal isolates was quantified using the Nesslerization method as described by Imada *et al.* [40]. Five mycelial discs, each 5 mm in diameter, were prepared from the margin of a seven–day–old colony and inoculated into 50 mL of MCD broth culture medium (pH 6.2) in a 100 mL Erlenmeyer flask. The flasks were incubated in a shaker incubator at 120 rpm and 28°C for five days. Then, the culture filtrate was obtained from the liquid culture medium using a Buchner funnel and vacuum pump. The obtained culture filtrate was centrifuged at 10000 rpm for 2 min, and the supernatant was considered the crude enzyme solution. The reaction mixture included 100 µL of 0.05 M Tris-HCl buffer (pH 7.2), 200 µL of 0.04 M L–asparagine (prepared in Tris-HCl buffer) and 100 µL of sterile distilled water (SDW). For test samples, 100 µL of crude enzyme was also added. The reaction mixture was vortexed and incubated at $37 \pm 2$°C for 1 h. To halt the reaction, 100 µL of 1.5 M trichloroacetic acid (TCA) was added, and the samples were kept at room temperature for 15 min. Blank samples received 100 µL of crude enzyme after the reaction was quenched. All samples (tests and blanks) were centrifuged at 10000 rpm at 4°C for 5 min, and 125 µL of the supernatant was transferred to a clean microtube as the enzyme mixture. For the colorimetric assay of L–asparaginase activity, 125 µL of Nessler's reagent along with 1 mL of SDW were also added to the microtube. The Nesslerization mixture was stabilized at room temperature for 20 min. The absorbance of the samples was measured at 450 nm using a UV–Visible spectrophotometer (Shimadzu, UV- 2501PC). The experiment was conducted in triplicate for both test and blank samples. The standard curve was generated using absorbance values of different concentrations of ammonium sulfate solution. One unit of L–asparaginase activity was defined as the amount of enzyme that catalyzes the cleavage of L–asparagine and releases 1 µmol of ammonium per minute at $37 \pm 2$°C [37,39]. L–asparaginase activity was calculated using the following equation [19,37,41]:

$$\text{Enzyme activity} \left(\text{Units mL}^{-1}\right) = \frac{\mu\text{mol of released NH}_4 \times V_1}{V_2 \times T \times V_3}$$

$V_1$ = the volume of initial reaction mixture in mL
$V_2$ = the volume of enzyme mixture used in final Nesslerization mixture in mL
$T$ = incubation time in minute
$V_3$ = total volume of crude enzyme used in initial reaction mixture and enzyme mixture used in final Nesslerization mixture in mL

## Effect of substrate on L–asparaginase activity

The effect of L–asparagine as the substrate on L–asparaginase activity of some selected endophytic fungal isolates was also evaluated using MCD broth culture medium with and without L–asparagine supplementation. The rest of the procedure was also performed using the above-mentioned method. The assays were conducted in triplicate [42,43].

## Effect of different culture media on L–asparaginase production

Ten different culture media were used including: Sucrose proline agar (g L$^{-1}$): sucrose 6, proline 2.7, K$_2$HPO$_4$ 1.3, KH$_2$PO$_4$ 1, KCl 5, MgSO$_4$·7H$_2$O 0.5, FeSO$_4$·7H$_2$O 0.01, ZnSO$_4$·7H$_2$O 0.002, MgCl$_2$ 0.0016, Mineral salts agar (g L$^{-1}$): KH$_2$PO$_4$ 0.7, K$_2$HPO$_4$ 0.7, MgSO$_4$·7H$_2$O 0.7, NH$_4$NO$_3$ 1, NaCl 0.005, FeSO$_4$·7H$_2$O 0.002, ZnSO$_4$·7H$_2$O 0.002, MnSO$_4$·7H$_2$O 0.001, Asthana and Hawker culture medium A (g L$^{-1}$): glucose 5, KNO$_3$ 3.5, KH$_2$PO$_4$ 1.75, MgSO$_4$·7H$_2$O 0.75, Elliott agar (g L$^{-1}$): dextrose 5, KH$_2$PO$_4$ 1.36, Na$_2$CO$_3$ 1.06, MgSO$_4$·7H$_2$O 0.5, asparagine 1, Brown agar (g L$^{-1}$): glucose 2, asparagine 2, K$_2$HPO$_4$ 1.25, MgSO$_4$·7H$_2$O 0.75, Dox agar (g L$^{-1}$): sucrose 7.5, KNO$_3$ 1, MgSO$_4$·7H$_2$O 0.25, K$_2$HPO$_4$ 0.5, FeCl$_2$ 0.025, Cerelose ammonium nitrate (g L$^{-1}$): cerelose (D–glucose) 50, NH$_4$NO$_3$ 10, KH$_2$PO$_4$ 5, MgSO$_4$·7H$_2$O 2.5, FeCl$_3$·6H$_2$O 0.02, Citrate agar culture medium (g L$^{-1}$): NaCl 5, MgSO$_4$·7H$_2$O 0.2, (NH$_4$)$_2$HPO$_4$ 1, K$_2$HPO$_4$ 1, sodium citrate

$(Na_3C_6H_5O_7)\cdot5.5H_2O$ 2.77, Kuehner basal culture medium (g L$^{-1}$): dextrose 20, asparagine 2, $KH_2PO_4$ 1.5, $MgSO_4\cdot7H_2O$ 0.5, $CaCl_2$ 0.33, $(NH_4)_2SO_4$ 2, KI 0.001, Piefer, Humphrey, and Acree medium (g L$^{-1}$): glucose 40, $K_2HPO_4$ 4, asparagine 4, $(NH_4)_2HPO_4$ 2, $MgSO_4\cdot7H_2O$ 2, $CaCO_3$ 0.25, $CaCl_2$ 0.1. Each culture medium was supplemented with 15 g L$^{-1}$ agar and modified by replacing the original nitrogen source with 10 g L$^{-1}$ L–asparagine. Thus, all culture media contained 10 g L$^{-1}$ L–asparagine as the sole nitrogen source. BTB (0.01%) was used as a pH indicator, and the final pH of all culture media was adjusted to 5.5. Twelve endophytic fungal isolates were grown on PDA culture medium for seven days. The plate assay was conducted using the above-mentioned method. The experiment was conducted in triplicate. The culture media without L–asparagine were used as the negative control [12,44,45].

## Statistical analysis

The statistical analysis of all data was performed using IBM SPSS Statistics software, version 26. The normality of the data and the homogeneity of variance were assessed using the Shapiro-Wilk and Levene's tests, respectively.All charts were generated using Microsoft Excel 2010. The heatmap chart was constructed using CIMminer oneMatrix tool developed by the National Cancer Institute [https://discover.nci.nih.gov/cimminer/oneMatrix.do].

## Results

### Endophytic fungi

In the present study, 62 endophytic fungal isolates that were previously obtained from five different plant species were screened for L–asparaginase production. These included 20 isolates from *Punica granatum*, 18 from *Pistacia vera*, 12 from *Prunus avium*, 10 from *Prunus cerasus* and 2 from *Taxus baccata* (Table 1).

### Qualitative assay of L–asparaginase production

The plate assay was performed to evaluate L–asparaginase production by endophytic fungi, as described by Gulati *et al.* [35]. Out of the 62 fungal isolates, 49 formed a blue–colored zone around the colony, indicating their potential to produce L–asparaginase (Fig 1). No color change was observed in the negative control. The zone index values ranged from 0.00 to 6.58 (S1 Fig). *Cladosporium perangustum* EL1 exhibited maximum color zone index (6.58). Thirteen fungal isolates did not produce L–asparaginase in this assay. One–way analysis of variance (ANOVA) revealed a significant difference (p–value $= 3.5 \times 10^{-87}$) among the fungal isolates in L–asparaginase production (Table 2). Duncan's post hoc test (p–value $= 3.5 \times 10^{-87}$) was applied to classify the fungal isolates based on their L-asparaginase production (S1 Fig).

### L–glutaminase and urease co–activities of L–asparaginase

The L–glutaminase and urease co-activities in L–asparaginase-producing isolates were evaluated using the conventional plate assay (Fig 1). The zone indexvalues for L–glutaminase production ranged from 0.00 to 3.89. *Cladosporium tenuissimum* ARA3 and *Nectria* sp. 4MOH exhibited the maximum L–glutaminase production. Nineteen isolates did not produce this enzyme (S2 Fig). The zone index values for urease production ranged from 0.41 to 16.67. *Cladosporium tenuissimum* ARA2 exhibited the maximum urease production, whereas four isolates — *Acremonium sclerotigenum* 8PK-2, *Neoscytalidium dimidiatum* URA1, *Neoscytalidium dimidiatum* I42, and *Neoscytalidium novaehollandiae* KhDS2–3 — showed the minimum zone index values (S3 Fig). One-way ANOVA revealed significant differences (p < 0.05) among the fungal isolates in L-glutaminase (p–value $= 1.4 \times 10^{-49}$) and urease (p–value $= 1.0 \times 10^{-108}$) production (Table 2). Duncan's post hoc test (p < 0.05) was applied to classify the fungal isolates based on their L-glutaminase and urease production (S2 and S3 Figs). The zone index values for L–asparaginase, L–glutaminase, and urease production in L-asparaginase-producing fungal isolates are presented in Fig 2. Moreover, the eleven best–performing endophytic fungal isolates were compared through heatmap analysis (Fig 3).

**Table 1. Origin and zone indices of L–asparaginase, L–glutaminase, and urease production of the endophytic fungal isolates used in this study.**

| Isolate | Taxon | Host | Zone index | | |
|---------|-------|------|------------|--|--|
| | | | ASNase | GLNase | urease |
| G88 | *Cytospora leucostoma* | *Taxus baccata* L. | – | n | n |
| KK4 | anamorphic *Xylaria* | | 2.32 | 1.14 | 1.68 |
| I11 | *Chaetomium globosum* | *Pistacia vera* L. | 0.40 | – | 2.55 |
| I48 | *Fusarium chlamydosporum* | | 1.75 | 1.54 | 1.59 |
| I16 | *Alternaria malorum* | | 2.22 | – | 2.84 |
| I6 | *Cytospora chrysosperma* | | – | n | n |
| A | *Ulocladium* sp. | | 1.90 | – | 2.27 |
| B | *Scopulariopsis brevicaulis* | | 1.70 | 2.18 | 2.27 |
| C | *Alternaria brassicae* | | 3.49 | – | 1.97 |
| I34 | *Nigrospora oryzae* | | 1.87 | 2.04 | 2.47 |
| I43 | *Acremonium sclerotigenum* | | 3.33 | 2.92 | 4.40 |
| I9 | *Chaetomium elatum* | | – | n | n |
| D | *Byssochlamys nivea* | | – | n | n |
| I42 | *Neoscytalidium dimidiatum* | | 0.30 | – | 0.41 |
| I27 | *Aspergillus tamarii* | | 2.01 | 1.07 | 1.20 |
| I10 | *Aspergillus flavus* | | 2.07 | 1.71 | 1.78 |
| I4 | *Aspergillus nidulans* | | 2.14 | 1.68 | 1.74 |
| E | *Penicillium chrysogenum* | | 2.88 | 2.88 | 3.54 |
| I46 | *Trichoderma longibrachiatum* | | 0.23 | 0.38 | 1.00 |
| F | *Aspergillus niger* | | – | n | n |
| 4MOH | *Nectria* sp. | *Prunus avium* L. | 3.45 | 3.78 | 4.56 |
| 11SA21 | *Acremonium egyptiacum* | | 1.91 | 1.13 | 1.48 |
| Ka14 | *Trichothecium roseum* | | 1.74 | 1.59 | 2.29 |
| CH18 | *Fusarium fujikuroi* | | 2.26 | 1.55 | 1.56 |
| KR9 | *Fusarium fujikuroi* | | 2.29 | 1.80 | 1.72 |
| 1M1 | *Chalastospora gossypii* | | 2.08 | 1.81 | 1.61 |
| TE33 | *Alternaria multiformis* | | 0.52 | – | 2.15 |
| 11SA31 | *Coniolariella limonispora* | | 4.18 | 2.87 | 4.00 |
| EL1 | *Cladosporium perangustum* | | 6.58 | 2.20 | 4.12 |
| GA11 | *Alternaria multiformis* | | 1.68 | – | 2.47 |
| KJ19 | *Alternaria tenuissima* | | 1.79 | – | 1.81 |
| 8PK-2 | *Acremonium sclerotigenum* | | 0.53 | 0.39 | 0.66 |
| ZA10 | *Lecanicillium muscarium* | *Prunus cerasus* L. | – | n | n |
| HAA10 | *Kalmusia variispora* | | – | n | n |
| SAA10 | *Acremonium egyptiacum* | | 2.22 | 1.88 | 1.70 |
| GLA1 | *Botryosphaeria dothidea* | | 0.82 | 0.74 | 1.03 |
| ESA1 | *Acremonium sclerotigenum* | | 2.12 | – | 2.64 |
| ARA3 | *Cladosporium tenuissimum* | | 4.66 | 3.89 | 8.61 |
| ARA2 | *Cladosporium tenuissimum* | | 5.90 | 2.30 | 16.67 |
| URA1 | *Neoscytalidium dimidiatum* | | 0.90 | – | 0.56 |
| ESA6 | *Alternaria rosea* | | 2.19 | – | 1.98 |
| ESA2 | *Alternaria rosea* | | 1.49 | – | 2.36 |

*(Continued)*

**Table 1.** (Continued)

| Isolate | Taxon | Host | Zone index | | |
|---|---|---|---|---|---|
| | | | ASNase | GLNase | urease |
| mr7−1 | *Fusarium prolifratum* | *Punica granatum* L. | 2.38 | 1.18 | 1.51 |
| YY1−1 | *Fusarium fujikuroi* | | 2.59 | 1.96 | 1.81 |
| KhlZS1−4 | *Fusarium solani* | | 2.43 | 1.31 | 2.00 |
| EES3−1 | *Bipolaris sorokiniana* | | 2.22 | − | 2.75 |
| Zn8−2 | *Sporormiella australis* | | − | n | n |
| IH1−2 | *Aureobasidium pullulans* | | 4.87 | − | 5.23 |
| EES2−2 | *Chaetomium globosum* | | − | n | n |
| KhlZB4−8 | *Cladosporium herbarum* | | 5.34 | 1.85 | 6.08 |
| Kr5−2 | *Cladosporium cladosporioides* | | 4.72 | 3.08 | 3.70 |
| KhDS2−3 | *Neoscytalidium novaehollandiae* | | 0.75 | − | 0.41 |
| YAR3−2 | *Aureobasidium melanogenum* | | − | n | n |
| KR9−1 | *Aureobasidium pullulans* | | − | n | n |
| mr3−6 | *Alternaria atra* | | 1.85 | − | 3.49 |
| mRa-2 | *Pelectosphaerella cucumerina* | | 3.63 | 1.54 | 3.45 |
| KhMJS-2 | *Cytospora punicae* | | − | n | n |
| YTa1-1 | *Purpureocillium lilacinum* | | − | n | n |
| IH2−7 | *Alternaria gaisen* | | 2.01 | − | 1.72 |
| EES3−2 | *Bipolaris sorokiniana* | | 1.68 | − | 2.74 |
| Kr4−1 | *Trichothecium roseum* | | 1.35 | 2.11 | 4.81 |
| IIV3−3 | *Alternaria longipes* | | 2.23 | − | 1.75 |

−: lack of enzyme production, n: without plate assay.

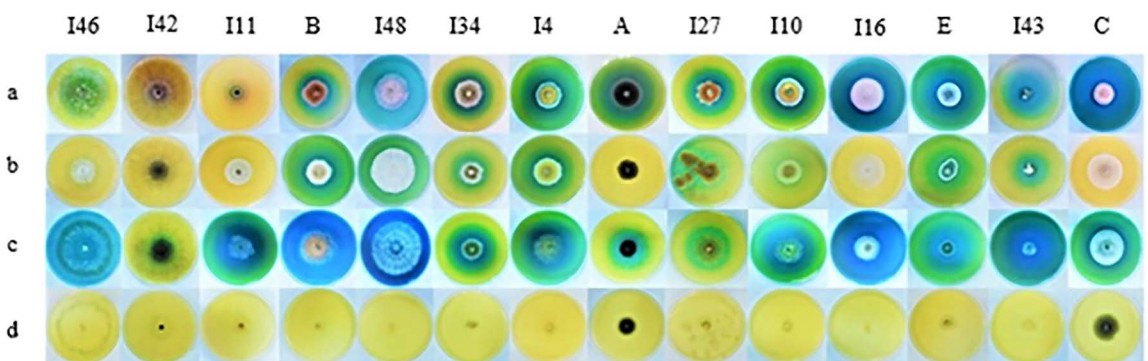

**Fig 1. Screening of endophytic fungal isolates for L−asparaginase, L−glutaminase, and urease production on MCD agar culture medium supplemented with bromothymol blue (BTB) after five days of incubation at 25°C under continuous dark conditions.** Labels "a," "b," and "c" indicate MCD agar supplemented with L−asparagine, L−glutamine, and urea as the sole nitrogen sources, respectively. Label "d" represents isolates grown on MCD agar culture medium without any nitrogen source.

## Quantitative assay of L−asparaginase activity

Twelve fungal isolates (*Cytospora leucostoma* G88, *Neoscytalidium novaehollandiae* KhDS2−3, *Sporormiella australis* Zn8−2, *Chaetomium globosum* EES2−2, *Cladosporium perangustum* EL1, *Neoscytalidium dimidiatum* URA1, *Aspergillus tamarii* I27, *Alternaria longipes* IIV3−3, *Acremonium egyptiacum* SAA10, *Aureobasidium pullulans* IH1−2, *Alternaria*

**Table 2. One–way ANOVA results indicated significant differences (p < 0.05) among fungal isolates in the production of L–asparaginase, L–glutaminase, and urease. All experiments were cunducted in triplicate. The sample size for L–asparaginase production consisted of 62 isolates and for both L–glutaminase and urease production was 49 isolates.**

| | | Sum of Squares | df | Mean Square | F | Sig. |
|---|---|---|---|---|---|---|
| **L–asparaginase** | Between Groups | 464.401 | 61 | 7.613 | 231.599 | $3.5 \times 10^{-87}$ |
| | Within Groups | 4.076 | 124 | 0.033 | | |
| | Total | 468.477 | 185 | | | |
| **L–glutaminase** | Between Groups | 187.900 | 48 | 3.915 | 143.995 | $1.4 \times 10^{-49}$ |
| | Within Groups | 2.664 | 98 | 0.027 | | |
| | Total | 190.564 | 146 | | | |
| **Urease** | Between Groups | 928.297 | 48 | 19.340 | 613.228 | $1.0 \times 10^{-108}$ |
| | Within Groups | 3.091 | 98 | 0.032 | | |
| | Total | 931.387 | 146 | | | |

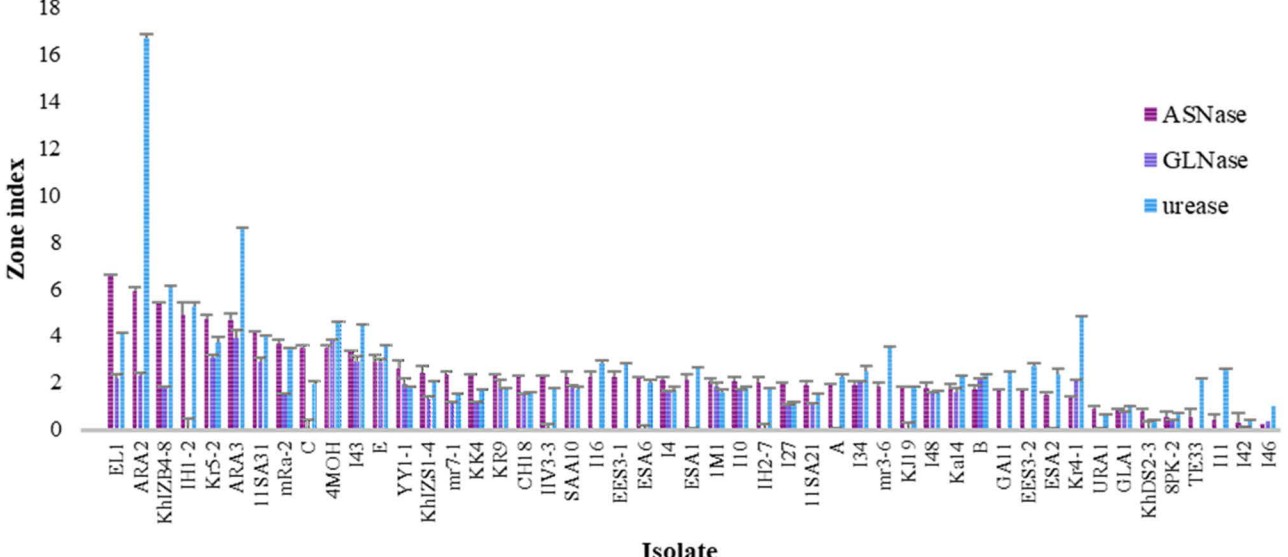

**Fig 2. L–asparaginase-producing endophytic fungal isolates and their L–glutaminase and urease co–activities.** Columns represent the mean zone indices for L–asparaginase, L–glutaminase, and urease after five days of incubation at 25°C under continuous dark conditions on MCD culture medium supplemented with L–asparagine, L–glutamine, and urea. Error bars indicate the standard error.

*brassicae* C, and *Cladosporium cladosporioides* Kr5–2) were selected for quantitative assay of L–asparaginase activity following the method described by Imada *et al.* [40], with slight modifications. The regression equation (y = 0.4767x) and the corresponding R–square (0.9814) were determined from the generated standard curve. An R² value close to 1 indicates that the regression model is appropriate for estimating the amount of ammonia released during enzyme activity (Fig 4).

The enzyme activity ranged from 0.56 to 10.78 U mL⁻¹. *Cladosporium cladosporioides* Kr5–2 (10.78 U mL⁻¹) and *Cytospora leucostoma* G88 (0.56 U mL⁻¹) exhibited the maximum and minimum enzyme activity, respectively. One-way ANOVA revealed a significant difference (p-value = $3.0 \times 10^{-12}$) among the selected fungal isolates in L–asparaginase activity in submerged fermentation (Table 3). Duncan's post hoc test (p–value = $3.0 \times 10^{-12}$) was applied to classify the fungal isolates (Fig 5).

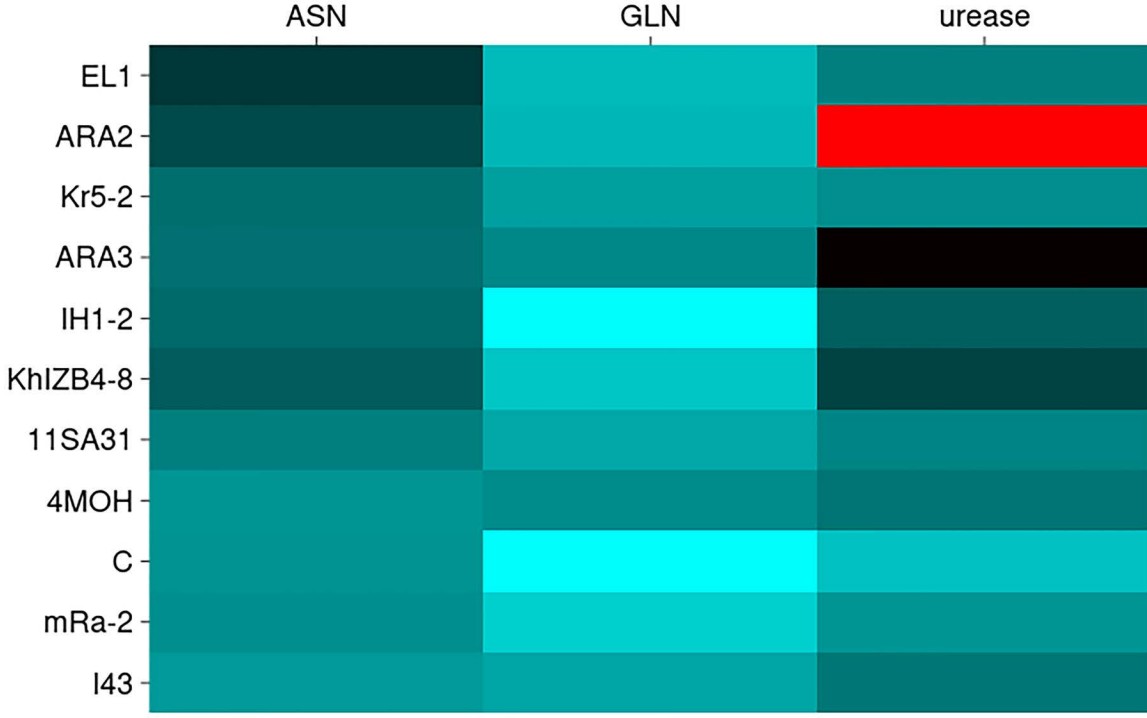

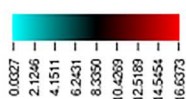

**Fig 3. Heatmap of best–performing endophytic fungal isolates for L–asparaginase production.** According to the heatmap, among the 11 endo-phytic fungal isolates that produced high levels of L–asparaginase, *Alternaria brassicae* C and *Aureobasidium pullulans* IH1–2 showed no L–glutami-nase production. In addition, *A. brassicae* C produced less urease compared to *Au. pullulans* IH1–2.

To assess the correlation between enzyme activity in submerged fermentation and enzyme production on solid cul-ture medium, the Pearson correlation coefficient was calculated. The results indicated a significant positive correlation (p-value = 0.016) between enzyme activity in submerged fermentation and enzyme production on solid culture medium. For the two–tailed correlation, the sample size (n), Pearson correlation coefficient (r), and p–value were 12, 0.674, and 0.016, respectively (Table 4). The correlation was visualized by constructing a scatterplot. The regression equation (y = 0.5209x − 0.0944) and corresponding $R^2$ value (0.4539) were determined (Fig 6).

### Effect of substrate on L–asparaginase activity

Out of the twelve selected endophytic fungal isolates, five (*Cytospora leucostoma* G88, *Sporormiella australis* Zn8–2, *Cladosporium perangustum* EL1, *Aureobasidium pullulans* IH1–2, and *Cladosporium cladosporioides* Kr5–2) were further used to evaluate the effect of L–asparagine as a substrate on L–asparaginase activity. The enzyme activity in MCD culture medium without L–asparagine ranged from 0.25 to 0.58 U mL$^{-1}$ (Fig 7). One-way ANOVA showed no significant differ-ences (p-value = 0.184) in enzyme activity among the fungal isolates in MCD broth culture medium without L–asparagine.

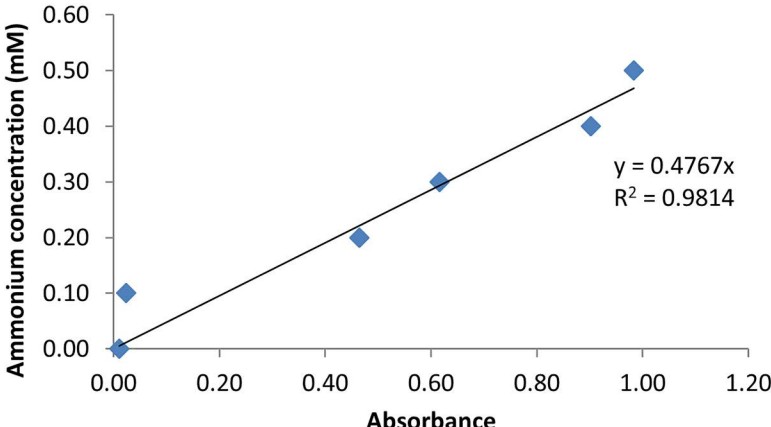

**Fig 4. Standard curve constructed using various concentrations of ammonium sulfate solution.** The regression equation and R² value are displayed within the plot area. An R² vale close to 1 indicates that the regression model is appropriate for estimating the amount of ammonia released during enzyme activity.

**Table 3. One–way ANOVA results indicated a significant difference (p-value = $3.0 \times 10^{-12}$) among fungal isolates in L–asparaginase activity. The sample size consisted of 12 endophytic fungal isolates and the experiment was conducted in triplicate.**

|  |  | Sum of Squares | df | Mean Square | F | Sig. |
|---|---|---|---|---|---|---|
| L-asparaginase activity | Between Groups | 263.696 | 11 | 23.972 | 90.306 | $3.0 \times 10^{-12}$ |
|  | Within Groups | 6.371 | 24 | 0.265 |  |  |
|  | Total | 270.067 | 35 |  |  |  |

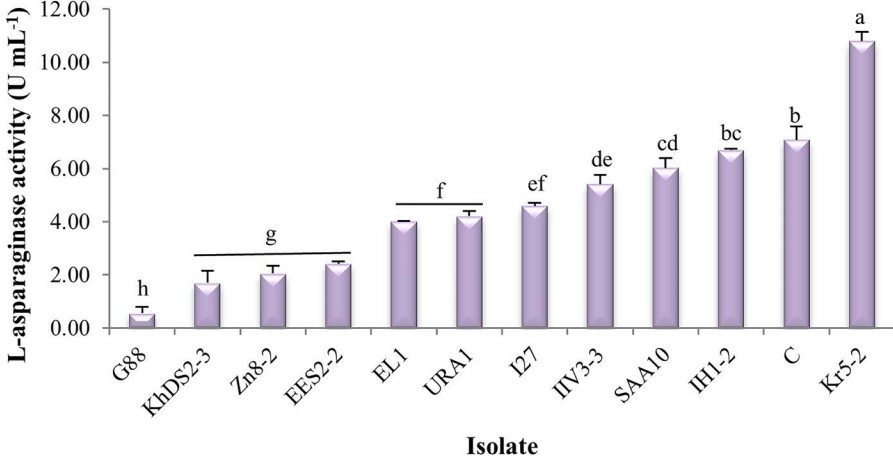

**Fig 5. L–asparaginase activity of selected endophytic fungal isolates under the submerged fermentation conditions.** Isolates were grouped using Duncan's post hoc test at a significance level of 0.05 (p–value = $3.0 \times 10^{-12}$). Letters above the columns indicate grouping; isolates sharing the same letter belong to the same statistical group. Error bars represent the standard error.

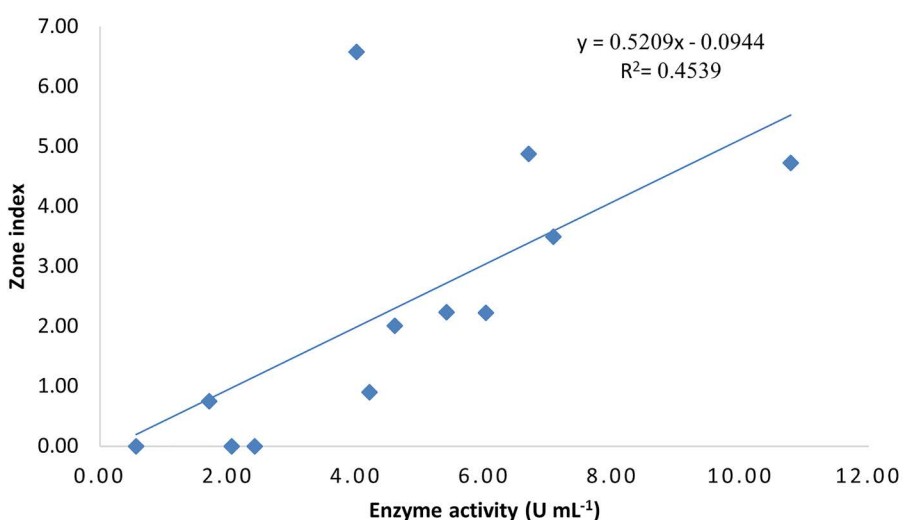

**Table 4. Result of the Pearson correlation test. Pearson correlation close to 1 and sig. (2–tailed) ≤ 0.05 (p–value = 0.016) displayed a significant strong direct correlation between zone index in solid culture media and enzyme activity in submerged fermentation.**

**Correlations**

|  |  | Enzyme activity | Zone index |
|---|---|---|---|
| Enzyme activity | Pearson Correlation | 1 | 0.674* |
|  | Sig. (2-tailed) |  | 0.016 |
|  | N | 12 | 12 |
| Zone index | Pearson Correlation | 0.674* | 1 |
|  | Sig. (2-tailed) | 0.016 |  |
|  | N | 12 | 12 |

*: indicates that correlation is significant at the 0.05 level (2–tailed).

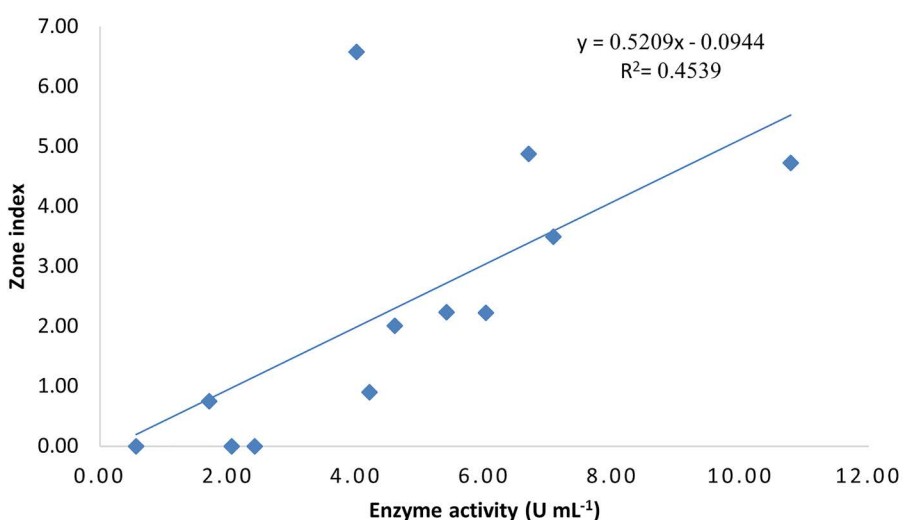

**Fig 6. Scatterplot of correlation between zone index in solid culture media and enzyme activity in submerged fermentation.** The regression equation and corresponding R² value are displayed in the plot erea.

In contrast, a significant difference (p–value = $1.6 \times 10^{-10}$) was observed when L–asparagine was added to the MCD culture medium (Table 5).

The effect of L–asparagine on L–asparaginase activity was assessed using an independent samples t–test. The results showed that L–asparagine had a significant effect on the enzyme activity of all tested isolates, with the exception of *Cytospora leucostoma* G88 (df = 4, p–value = 0.303) (Fig 7, S2 Table).

## Effect of different culture media on L–asparaginase production

The L–asparaginase production of 12 selected isolates was evaluated in ten different solid culture media using the conventional plate assay (Fig 8). One–way ANOVA revealed a significant difference (p < 0.05) in enzyme production among fungal isolates in each culture medium (S3 Table). Duncan's post hoc test (p < 0.05) was applied to classify the fungal isolates in each culture media (Fig 9).

The zone index in Sucrose proline agar culture medium ranged from 0.00 to 7.09. *Aureobasidium pullulans* IH1–2 (7.09) exhibited the maximum L–asparaginase production. Two isolates, *Sporormiella australis* Zn 8–2 and *Cytospora*

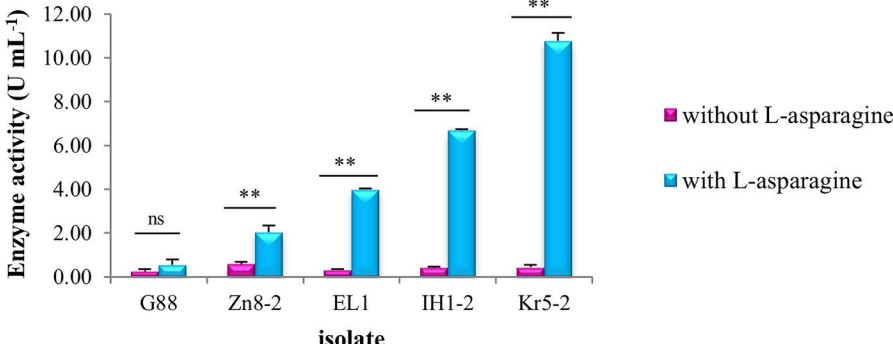

**Fig 7. L–asparaginase activity of examined endophytic fungal isolates in MCD culture medium with and without L–asparagine.** The sample size consisted of five endophytic fungal isolates. Each experiment was performed in triplicate. "ns" indicates no significant difference; "**" denotes a significant difference (p < 0.01) between enzyme activities in culture media with and without L–asparagine, based on an independent samples t-test. The exact p–value of enzyme activity in each isolate was determined in the ANOVA table (S2 Table). Error bars represent the standard error.

**Table 5. One–way ANOVA results showed no significant difference (p–value = 0.184) in L–asparaginase activity among fungal isolates cultured in MCD culture medium without L–asparagine as the sole nitrogen source. However, a significant difference (p–value = 1.6 × 10⁻¹⁰) was observed among the isolates when L–asparagine was supplemented into MCD culture medium.**

|  |  | Sum of Squares | df | Mean Square | F | Sig. |
|---|---|---|---|---|---|---|
| **Enzyme activity in MCD Culture medium with L–Asparagine** | Between Groups | 196.619 | 4 | 49.155 | 304.238 | $1.6 \times 10^{-10}$ |
|  | Within Groups | 1.616 | 10 | 0.162 |  |  |
|  | Total | 198.235 | 14 |  |  |  |
| **Enzyme activity in MCD Culture medium without L–asparagine** | Between Groups | 0.186 | 4 | 0.046 | 1.916 | 0.184 |
|  | Within Groups | 0.242 | 10 | 0.024 |  |  |
|  | Total | 0.428 | 14 |  |  |  |

*leucostoma* G88, did not produce L–asparaginase in this culture medium (Fig 6A). The zone index in Mineral salts agar culture medium ranged from 0.95 to 10.00. *Aureobasidium pullulans* IH1–2 (10.00) and *Cytospora leucostoma* G88 (0.95) exhibited the maximum and minimum L–asparaginase production, respectively. All tested isolates developed a color zone in this culture medium (Fig 6B). The zone index in Asthana and Hawker culture medium A ranged from 0.00 to 5.72. *Aureobasidium pullulans* IH1–2 (5.72) exhibited the maximum L–asparaginase production. Three isolates — *Sporormiella australis* Zn 8–2, *Chaetomium globosum* EES2–2, and *Cytospora leucostoma* G88 — did not produce L–asparaginase in this culture medium (Fig 6C). The zone index in Elliott agar culture medium ranged from 0.00 to 7.81. *Aureobasidium pullulans* IH1–2 (7.81) exhibited the maximum L–asparaginase production. Two isolates – *Sporormiella australis* Zn 8–2 and *Cytospora leucostoma* G88 – did not produce L–asparaginase in this culture medium (Fig 6D). The zone index in Brown agar culture medium ranged from 0.00 to 8.71. *Cladosporium perangustum* EL1 (8.71) exhibited the maximum L–asparaginase production. *Cytospora leucostoma* G88 did not produce L–asparaginase in this culture medium (Fig 6E). The zone index in Dox agar culture medium ranged from 0.00 to 5.56. *Cladosporium perangustum* EL1 (5.56) exhibited the maximum L–asparaginase production. Five isolates did not develop any color zone in this culture medium (Fig 6F). The zone index in Cerelose ammonium nitrate culture medium ranged from 0.00 to 1.69. *Cladosporium cladosporioides* Kr5–2 (1.69) exhibited the maximum L–asparaginase production. Nine isolates did not produce L–asparaginase in this culture medium (Fig 6G). The zone index in Citrate agar culture medium ranged from 1.54 to 9.56. According to Duncan's post hoc test, *Cladosporium perangustum* EL1 (9.56) and *Aureobasidium pullulans* IH1–2 (9.36) exhibited the maximum L–asparaginase production. In contrast, *Neoscytalidium novaehollandiae* KhDS2–3 (1.56) and *Acremonium egyptiacum*

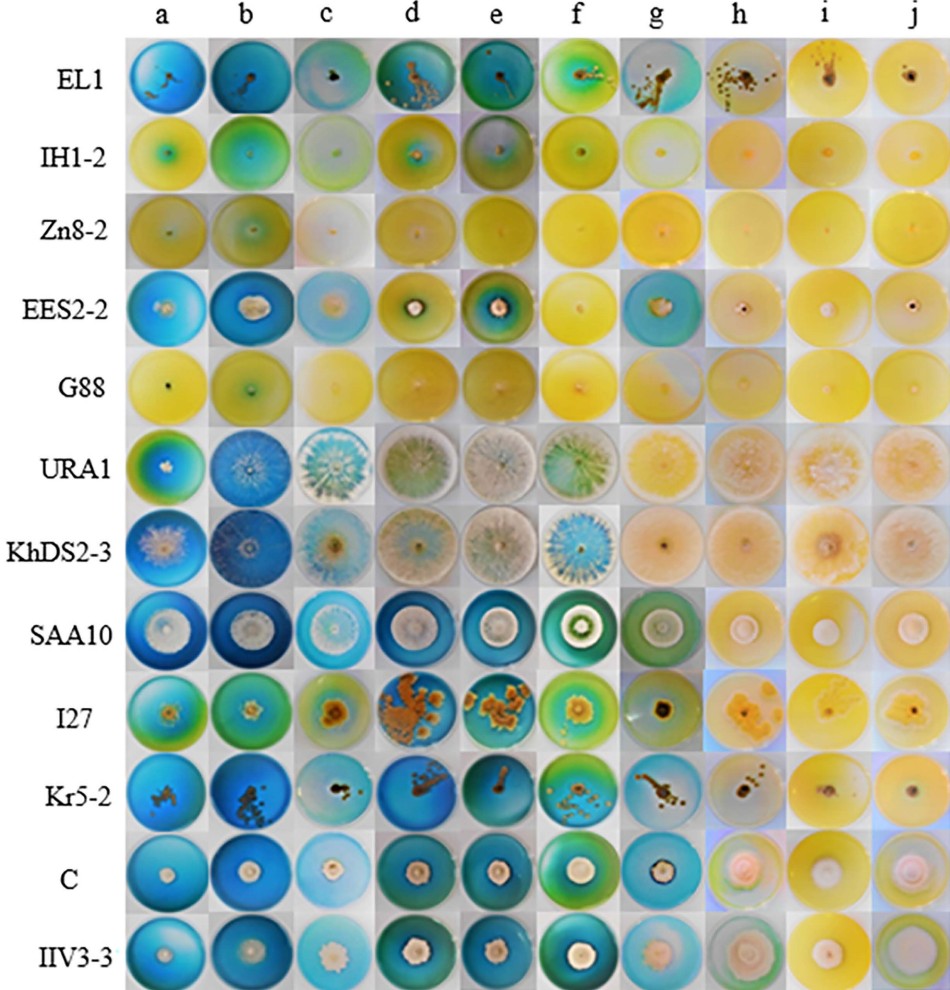

**Fig 8. L–asparaginase production by selected endophytic fungal isolates grown on ten different culture media.** Columns represent the culture media, labeled as follows: **a)** Citrate agar; **b)** Mineral salts agar; **c)** Brown agar; **d)** Sucrose proline agar; **e)** Elliott agar; **f)** Asthana and Hawker culture medium A; **g)** Dox agar; **h)** Kuehner basal culture medium; **i)** Cerelose ammonium nitrate agar; and **j)** Piefer, Humphrey, and Acree culture medium. Rows correspond to the fungal isolates.

SAA10 (1.54) exhibited the minimum enzyme production. All tested isolates developed a color zone in this culture medium (Fig 6H). The zone index in Kuehner basal culture medium ranged from 0.00 to 3.46. *Cladosporium cladosporioides* Kr5–2 (3.46) exhibited the maximum L–asparaginase production. Eight isolates did not produce L–asparaginase in this culture medium (Fig 6I). The zone index in Piefer, Humphrey, and Acree culture medium ranged from 0.00 to 2.48. *Alternaria longipes* IIV3–3 (2.48) exhibited the maximum L–asparaginase production. Nine isolates did not develop any color zone in this culture medium (Fig 6J). Moreover, based on the results of one–way ANOVA, L–asparaginase production showed a significant difference ($p < 0.05$) among culture media for each isolate (S4 Table). Enzyme production of each isolate in ten different culture media was compared to MCD as the basal culture medium using the Dunnett's post hoc test (S5 Table). The relationship between enzyme production of selected fungal isolates and tested culture media was visualized by constructing heatmap. Isolates and culture media were clustered using the average linkage clustering algorithm, and the heatmap was constructed using the Euclidean distance method (Fig 10).

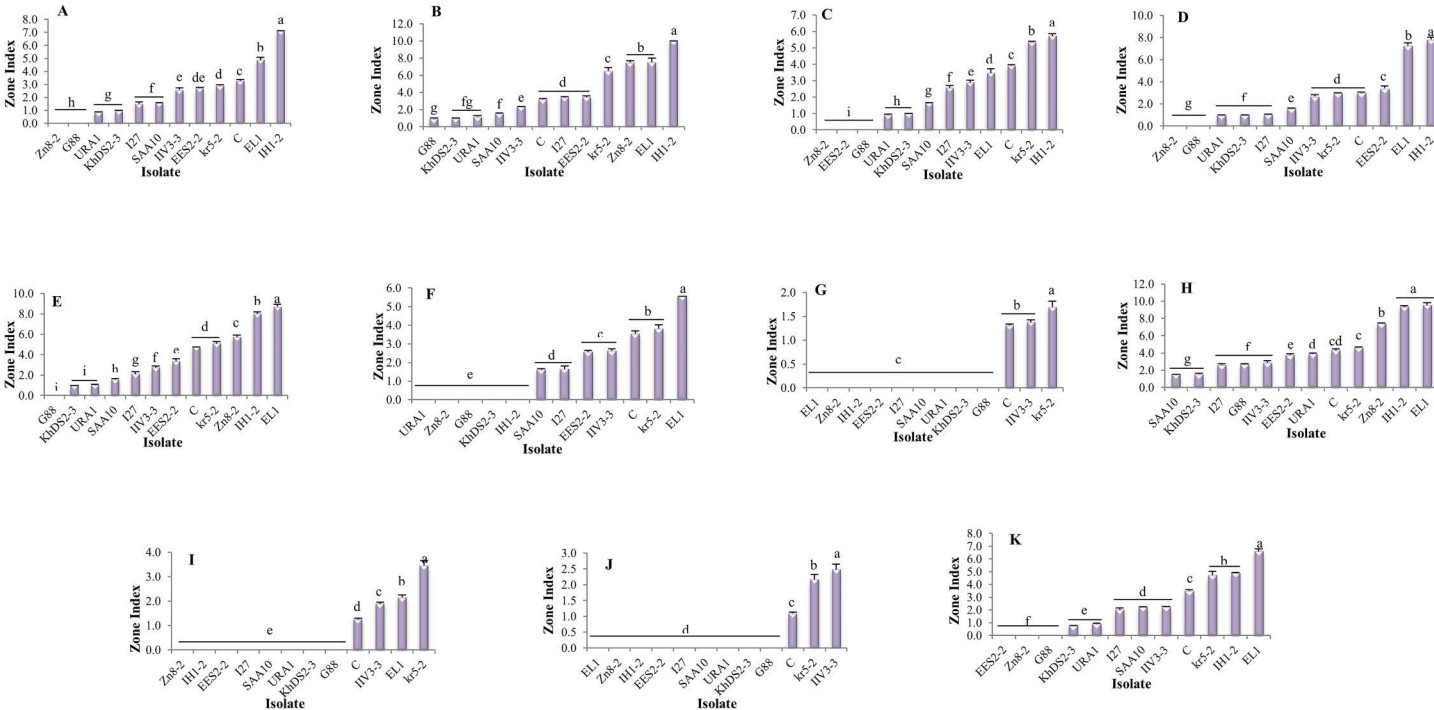

**Fig 9. Duncan's post hoc test of L–asparaginase production by selected endophytic fungi grown on ten different culture media at a significance level of 0.05: A) Sucrose proline agar (p–value = –3.0 × 10-36); B) Mineral salts agar (p–value = 3.0 × 10-30); C) Asthana and Hawker culture medium A (p–value = 3.0 × 10-33); D) Elliott agar (p–value = 2.0 × 10-33); E) Brown agar (p–value = 5.0 × 10-32); F) Dox agar (p–value = 1.0 × 10-31); G) Cerelose ammonium nitrate agar (p–value = 2.0 × 10-31); H) Citrate agar (p–value = 5.0 × 10-32); I) Kuehner basal culture medium (p–value = 1.0 × 1032); and J) Piefer, Humphrey, and Acree culture medium (p–value = 2.0 × 10-28). MCD (K) (p–value = 5.0 × 10⁻³²) was used as the basal culture medium. Each group indicated by a lowercase letter.** Isolates sharing the same letter belong to the same group. The sample size consisted of 12 endophytic fungal isolates and the experiments were performed in triplicate. Error bars represent the standard error.

## Discussion

L–asparaginase is a therapeutic enzyme used for the treatment of hematopoietic diseases, particularly acute lymphoblastic leukemia in children. This enzyme is also used in the food industry to mitigate acrylamide formation in starchy foods that are cooked at high temperatures. Endophytic fungi, which have a symbiotic relationship with their host plants, have attracted significant attention for their potential to produce therapeutic enzymes [46]. Studies have demonstrated that endophytic fungi can produce L–asparaginase without glutaminase and urease co-activities, resulting in an enzyme with reduce neurotoxicity and fewer side effects [4,37,47,48]. In this study, 62 endophytic fungal isolates from the culture collection were screened to identify a new source capable of producing L–asparaginase with no or low levels of glutaminase and urease co–activities. Forty-nine isolates were able to produce L–asparaginase, of which 19 were glutaminase-free. Among the fungi producing L–asparaginase without glutaminase activity, *Alternaria brassicae* C – characterized by a high L–asparaginase zone index (3.49) and low urease co-activity (1.97) – is recommended as a promising new eukaryotic source for producing L-asparaginase with fewer side effects. These results are consistent with previous studies. Yu *et al.* (2025) assessed the marine *Paraconiothyrium cyclothyrioides* strain MABIK FU00000820 for its ability to produce L–asparaginase free of L–glutaminase and urease co–activities using a conventional plate assay [15]. Luhana and Bariya (2025) identified *Aspergillus flavus* HK03 as a filamentous fungus capable of producing L–asparaginase free of L–glutaminase co–activity [49]. The results demonstrated that filamentous fungi can be considered as new sources for L–asparaginase production with fewer side effects. This isolate has also shown high enzyme activity (7.07 U mL⁻¹) in the

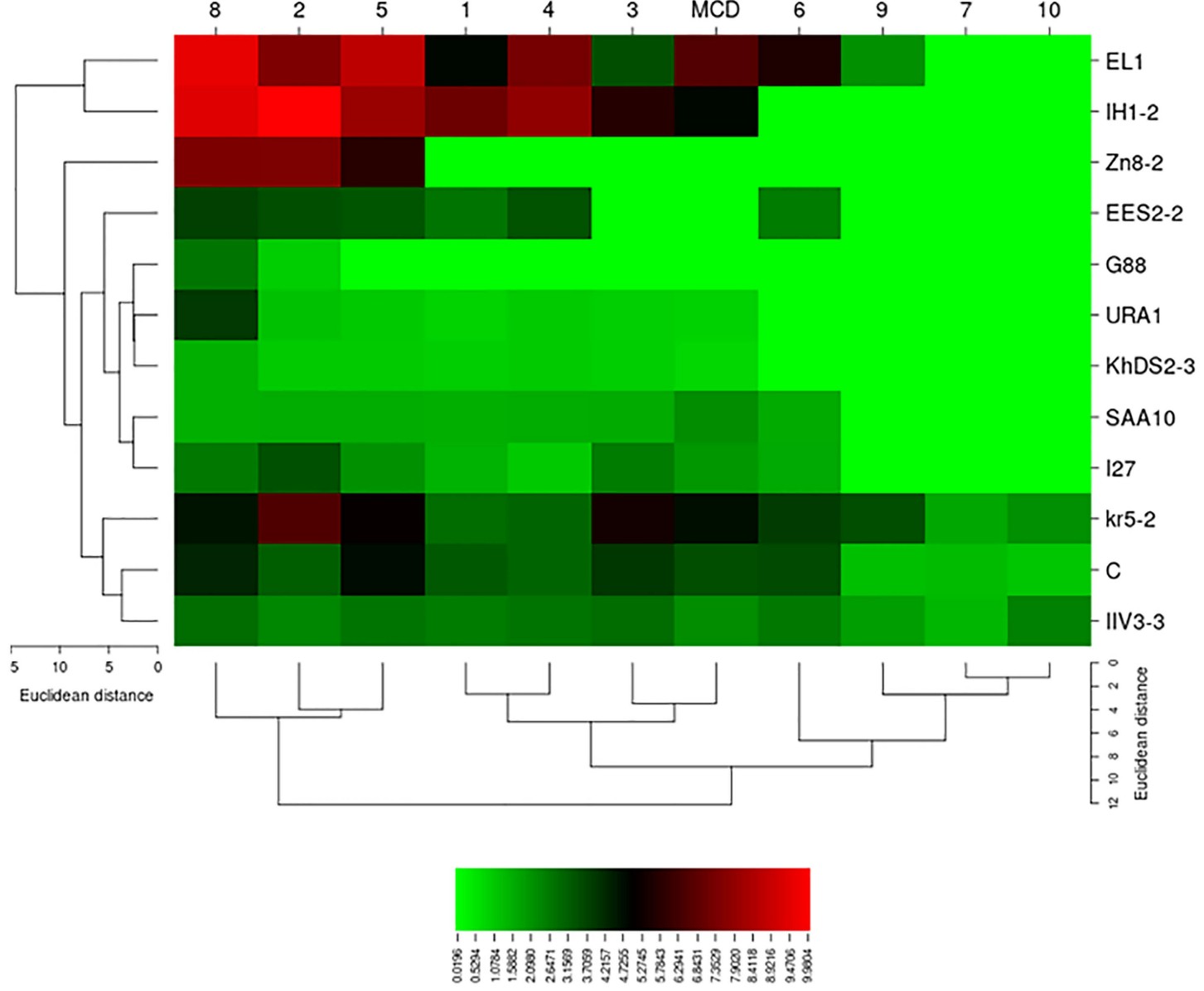

**Fig 10. Heatmap illustrating L–asparaginase production by endophytic fungal isolates on different culture media.** Columns represent the culture media, numbered 1 to 10 as follows: 1) Sucrose proline agar; 2) Mineral salts agar; 3) Asthana and Hawker culture medium A; 4) Elliott agar; 5) Brown agar; 6) Dox agar; 7) Cerelose ammonium nitrate agar; 8) Citrate agar; 9) Kuehner basal culture medium; and 10) Piefer, Humphrey, and Acree culture medium. MCD was used as the basal culture medium. Rows correspond to the tested fungal isolates.

quantification assay. In this study, 75% of fungal isolates were able to produce L–asparaginase, which is consistent with the findings of Chua *et al.* [39] (89.66%), Chakraborty and Shivakumar [38] (81%), Ashok *et al.* [36] (80%), de Andre *et al.* [4] (71%) and Hatamzadeh *et al.* [37] (45%). Hence, these results show that most fungal isolates are able to produce L–asparaginase. The ability of isolates from the culture collection in this study to produce L–asparaginase indicates the existence of the L–asparaginase-coding gene as well as its expression stability. Although almost a decade has passed since their isolation, the coding gene can still be effectively expressed. This result provides an advantage for these

isolates in long-term maintenance and utilization. To quantify and compare L–asparaginase production in solid and liquid culture media, 12 fungal isolates were selected from various Duncan's groups based on the plate assay. Three isolates that showed no L–asparaginase production in solid culture medium exhibited enzyme activity in submerged fermentation. *Neoscytalidium dimidiatum* URA1, which had a low zone index (0.09) in the plate assay, exhibited moderate enzyme activity (4.20 U mL$^{-1}$) in liquid fermentation; higher than *Cladosporium perangustum* EL (4.00 U mL$^{-1}$), which had the maximum zone index (6.58). *Aureobasidium pullulans* IH1–2, *Alternaria brassicae* C, and *Cladosporium cladosporioides* Kr5−2, which had high zone indices in the plate assay, also exhibited high enzyme activity in liquid culture media. In this study, *Cladosporium cladosporioides* Kr5–2 exhibited the maximum enzyme activity (10.78 U mL$^{-1}$), which was higher than *Fusarium proliferatum* Br08 (0.492 U mL$^{-1}$) [37] from the previous study conducted on endophytic fungi in Iran. It is hypothesized that the discrepancy between qualitative and quantitative assays is related to differences in the potential of fungi to secrete enzymes in solid and liquid culture media. This difference may be associated with physiological interactions between fungi and culture media. Moreover, the type of culture medium and the availability of the substrate can affect the metabolic pathways and enzyme production mechanisms in fungi. Despite the differences in enzyme production between solid and liquid culture media, our data analysis showed a strong and direct correlation between the zone index in solid culture medium and enzyme activity in submerged fermentation. This result is consistent with the findings of Gulati *et al.* [35]The current study also investigated the effect of L-asparagine as a substrate on enzyme activity in submerged fermentation. The enzyme activity of the five examined fungal isolates in liquid culture medium without L–asparagine remained at a similar level. However, the presence of L–asparagine led to a significant difference in enzyme activity in all isolates except *Cytospora leucostoma* G88. This isolate did not produce L–asparaginase in the plate assay and exhibited the lowest enzyme activity in both the culture medium supplemented with L–asparagine and the culture medium lacking the substrate. These results showed that although *Cytospora leucostoma* G88 carries the L–asparaginase–coding gene, it is inherently a poor producer of this enzyme and cannot be considered for industrial applications. The significant difference in enzyme activity between culture media with and without L–asparagine in the other four isolates suggests that although the absence of L–asparagine did not inhibit the expression of the L–asparaginase-coding gene, this extracellular enzyme is inducible and is under positive substrate regulation. These results are consistent with the findings of Udayan *et al.* (2023) and Sisay *et al.* (2024), who demonstrated that the nitrogen source significantly influences L–asparaginase activity, with the highest activity observed when L–asparagine was used as the sole nitrogen source [30,50]. This may be because L–asparagine is the primary substrate for L–asparaginase and plays a crucial role in various metabolic pathways. It seems that extracellular L–asparaginase functions as an auxiliary enzyme for the decomposition of secondary nitrogen sources in the environment. It is hypothesized that when an easily assimilable nitrogen source is not available for fungal isolates, the L–asparaginase production pathway is activated and induced by the available substrate in the culture medium. In the current work, endophytic fungi were screened for L–asparaginase production using the modified Czapek Dox agar culture medium and the conventional plate assay as described by Gulati *et al.* [35]. Quantitative assay revealed that some fungal isolates, despite exhibited no colored zone in solid culture medium, were able to produce L–asparaginase in submerged fermentation. Therefore, identifying an effective culture medium is essential for the rapid screening of a large fungal population for L–asparaginase production. Accordingly, an additional experiment was conducted to identify a more effective culture medium capable of revealing the full potential of endophytic fungi for L–asparaginase production within a five–day period.In this experiment, all nitrogen sources were omitted from the selected culture media and used L–asparagine as the sole nitrogen source to ensure that pathways involved in the degradation of alternative nitrogen sources were not activated. Out of ten selected solid culture media, Mineral salts agar and Citrate agar were the most suitable, and all tested fungal isolates developed a blue color zone indicating L–asparaginase production. L–asparaginase production in these culture media was higher than in MCD, the basal culture medium, indicating that these culture media enhanced induction of L–asparaginase. Evaluation the true potential for enzyme production in fungal isolates ensures that all isolates capable of producing L–asparaginase are considered. This is important because among these isolates, there

might be one with prominent enzyme activity in submerged fermentation or whose enzyme structure can be easily engineered, allowing its use for industrial applications and scale–up. The results of this study also show that the type of culture medium affects L–asparaginase production. For instance, 75% of the tested fungal isolates did not produce L–asparaginase on Cerelose ammonium nitrate agar and Piefer, Humphrey, and Acree culture medium. Similarly, L–asparaginase production was not observed in 66.7% of the tested isolates in the Kuehner basal culture medium. It is suggested that these results may be related to the high glucose concentration in these culture media. Previous studies have shown that high glucose concentration makes the culture medium more acidic and inhibits L–asparaginase production [4,12]. However, this hypothesis requires further experimental validation to evaluate the effect of glucose concentration on L–asparaginase production.

## Conclusions

In this study, 19 endophytic fungal isolates were able to produce L–asparaginase free of L–glutaminase co–activity. Among them, the isolate *Alternaria brassicae* C was proposed as the best endophytic fungal isolate, exhibiting the high L-asparaginase zone index, high enzyme activity, and low urease co–activity. The current study demonstrated a direct correlation between L–asparaginase production in solid and submerged culture media. Moreover, this study demonstrated that L–asparaginase is an inducible enzyme and is positively regulated by its substrate. This study also suggested that among the tested culture media, Mineral salts agar and Citrate agar were the most effective for L–asparaginase production and can be used in the L–asparaginase screening program. However, further *in vitro* and *in vivo* studies are required to replace commercial bacterial sources with a novel eukaryotic source of L–asparaginase production that has fewer side effects.

## Supporting information

**S1 Table. Characteristics of studied endophytic fungal isolates candidates (names, host sources, and accession numbers of ITS1–5.8S–ITS2 rDNA sequences).**
(PDF)

**S2 Table. Results of independent samples t–test analysis.** According to Levene's test, the variances between enzyme activity in the culture media with and without substrate are equal and independent two-tailed t–test is significant in studied isolates except in *Cytospora leucostoma* G88 (p–value = 0.303).
(PDF)

**S3 Table. One–way ANOVA results indicated a significant difference (p < 0.05) among fungal isolates in each culture media in L–asparaginase production.** The sample size consisted of 12 endophytic fungal isolates and the experiment was conducted in triplicate.
(PDF)

**S4 Table. One–way ANOVA results indicated a significant difference (p < 0.05) among culture media for L–asparaginase production in each isolate.** The sample size consisted of 11 culture media (MCD plus 10 additional culture media) and the experiment was conducted in triplicate.
(PDF)

**S5 Table. Dunnett's t–tests treat one group (MCD culture medium) as a control, and all other groups (ten different culture media) were compared against it.**
(PDF)

**S1 Fig. L–asparaginase production in 62 endophytic fungal isolates.** Mean zone index for each isolate is presented. Isolates were grouped based on Duncan's post hoc test at a significance level of 0.05 (p–value = $3.5 \times 10^{-87}$). Letters above

each column indicate the groups identified by Duncan's analysis; isolates sharing the same letter belong to the same group. Error bars represent the standard error.
(TIF)

**S2 Fig. L–glutaminase production in 49 isolates of L–asparaginase–producing endophytic fungi.** Mean zone index for each isolate is presented. Isolates were grouped using Duncan's post hoc test at a significance level of 0.05 (p-value = $1.4 \times 10^{-49}$). Letters above each column indicate groupings based on Duncan's analysis; isolates sharing the same letter belong to the same group. Error bars represent the standard error.
(TIF)

**S3 Fig. Urease production in 49 isolates of L–asparaginase-producing endophytic fungi.** Mean zone index for each isolate is presented. Isolates were grouped using Duncan's post hoc test at a significance level of 0.05 (p-value = $1.0 \times 10^{-108}$). Letters above each column indicate groupings based on Duncan's analysis; isolates sharing the same letter belong to the same group. Error bars represent the standard error.
(TIF)

## Acknowledgments

The research has been performed in the University of Tehran, so we are grateful to the University of Tehran.

## Author contributions

**Formal analysis:** Zahra Zaeimian.

**Methodology:** Khalil-Berdi Fotouhifar, Mohsen Farzaneh.

**Resources:** Mohsen Farzaneh.

**Supervision:** Khalil-Berdi Fotouhifar.

**Writing – original draft:** Zahra Zaeimian.

**Writing – review & editing:** Khalil-Berdi Fotouhifar.

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
