## [Decision Letter · Decision Letter 0]

27 Oct 2025

Dear Dr. Fotouhifar,

Thank you for submitting your manuscript to PLOS ONE. After careful consideration, we feel that it has merit but does not fully meet PLOS ONE’s publication criteria as it currently stands. Therefore, we invite you to submit a revised version of the manuscript that addresses the points raised during the review process.

We look forward to receiving your revised manuscript.

Kind regards,

Shashi Kant Bhatia

Academic Editor

PLOS ONE

Journal Requirements:

2. We note that your Data Availability Statement is currently as follows: All relevant data are within the manuscript and in Supporting Information files.

Reviewers' comments:

Reviewer's Responses to Questions

**Comments to the Author**

1. Is the manuscript technically sound, and do the data support the conclusions?

Reviewer #1: Yes

Reviewer #2: Yes

Reviewer #3: Partly

Reviewer #4: No

2. Has the statistical analysis been performed appropriately and rigorously?

Reviewer #1: Yes

Reviewer #2: Yes

Reviewer #3: Yes

Reviewer #4: Yes

3. Have the authors made all data underlying the findings in their manuscript fully available?

Reviewer #1: Yes

Reviewer #2: Yes

Reviewer #3: No

Reviewer #4: Yes

4. Is the manuscript presented in an intelligible fashion and written in standard English?

Reviewer #1: Yes

Reviewer #2: Yes

Reviewer #3: Yes

Reviewer #4: Yes

Reviewer #1: Greetings, good work but kindly edit the following minor points below:

1. In this manuscript, the pronoun "We" (9 times) was used. In scientific writing, it is better to avoid the pronouns. Please replace them with formal scientific expressions such as "This study," "The present study," or "The current study."

2-Several references [4,13,16, 17, 23, 32, 33, 34, 35, 36, 38, 42, 43, 45, 46, 50] are old. Please try to cite more recent references.

Kind regards

Reviewer #2: Dear author,

The article "L-asparaginase activity in some endophytic fungi: glutaminase-free and low urease coactivities" was reviewed. Please make the following corrections:

1- References 36, 42, 46, and 50 are old, please use new references.

2- Please insert the suitable footnotes for Figures 1-7.

Kind regards

Reviewer #3: The manuscript entitled “L-asparaginase activity in some endophytic fungi glutaminase-free and low urease co-activities” presents an interesting investigation into the screening of endophytic fungi for L-asparaginase production with minimal co-activities. The topic is relevant to both clinical enzyme research and industrial biotechnology. Overall, the study appears technically sound, but several aspects require clarification and improvement before it can be considered for publication.

1. General Assessment

The paper is generally well written in clear English and organized logically. The rationale of identifying new fungal sources of L-asparaginase is scientifically valid and within the scope of PLOS ONE. However, the manuscript would benefit from more detailed methodological descriptions, improved data transparency, and a clearer articulation of novelty.

2. Methodology

The isolation sources and strain identifiers of the fungal isolates should be more explicitly described to ensure reproducibility.

The parameters of the Nesslerization assay (calibration curve, regression equation, and R² value) should be presented.

Statistical methods (ANOVA, Duncan’s test, t-test) are appropriate but not fully reported. Please specify the sample size (n), number of replicates, degrees of freedom, and exact p-values.

Include a short statement on how data normality and variance homogeneity were verified before using parametric tests.

3. Data Availability

The manuscript claims that “all relevant data are within the manuscript and its supporting information,” but the raw data (enzyme activity values, absorbance readings, and statistical outputs) are not included. To comply with the PLOS Data Policy, the authors should upload all underlying datasets as supplementary files or in a public repository.

4. Results and Presentation

Figures are informative but should include error bars, axis labels, and units.

A summary table or graphical heatmap of the best-performing isolates would improve data clarity.

Correlation results should be visualized with a scatterplot.

Some numerical results (e.g., enzyme activity values) are repeated excessively; consider simplifying.

5. Discussion and Interpretation

The discussion effectively links results to previous studies but occasionally becomes descriptive rather than analytical.

The statement about glucose concentration inhibiting enzyme activity is speculative; it should be supported by data or moved to “future work.”

The claim that Alternaria brassicae C is a promising industrial strain is reasonable but needs further biochemical or kinetic validation.

6. Language and Formatting

The manuscript is written in standard, intelligible English, though minor stylistic corrections (sentence length, punctuation, and consistency of scientific names) are recommended. Please italicize all genus and species names and ensure consistent use of units (e.g., U mL⁻¹).

7. Ethical and Publication Standards

No ethical or publication misconduct concerns were identified. The authors clearly stated that no competing interests or external funding exist.

Summary Recommendation

The manuscript contains scientifically valid and relevant data but requires major revision to strengthen methodological transparency, data accessibility, and discussion depth. After appropriate revision, it could be suitable for publication in PLOS ONE.

Reviewer #4: The manuscript titled L-asparaginase activity in some endophytic fungi:glutaminase-free and low urease co-activities submitted to the Journal has very preliminary findings containing based on only zone clearance and media optimization. There are obvious conclusion made which can not be validated based on the current results. For example, the author claims that the best isolate has low urease activity which is almost 28% of the total activity. They concluded that Alternaria L-asparaginase is best one and can be used for commercial production but failed to show for what commercial purpose. Eventually the ultimate final end goal is missing. Merely reporting isolation of L-asparaginase production from endophytic fungi can not solve the purpose. There is no comparision of production parameters, size of the enzyme (need to go beyond qualtitative tests). In my opinion, the work is too premature to be published at this point.

**Do you want your identity to be public for this peer review?** For information about this choice, including consent withdrawal, please see our Privacy Policy

Reviewer #1: No

Reviewer #2: No

Reviewer #3: No

Reviewer #4: No

---

## [Author Response · Author response to Decision Letter 1]

10 Dec 2025

Reviewer #1:

1. In this manuscript, the pronoun "We" (9 times) was used. In scientific writing, it is better to avoid the pronouns. Please replace them with formal scientific expressions such as "This study," "The present study," or "The current study."

Response: Your comment has been addressed in the manuscript and corrected as you suggested.

2-Several references [4,13,16, 17, 23, 32, 33, 34, 35, 36, 38, 42, 43, 45, 46, 50] are old. Please try to cite more recent references.

Response: References 32, 33, 34, and 35 are those to which the studied endophytic fungal isolates belong, so they cannot be replaced. References 36, 42, 45, and 46 describe the main methods (qualitative and Nesslerization assays) and the culture media used, and therefore cannot be replaced. The other outdated references have been removed or replaced with new ones.

Reviewer #2:

1- References 36, 42, 46, and 50 are old, please use new references.

Response: References 36, 42, and 46 are the most important sources on which the qualitative assay, Nesslerization method, and some of the culture media used in the present study are based, and therefore cannot be replaced. Reference 50 has been removed in the revised version.

2- Please insert the suitable footnotes for Figures 1-7.

Response: For all figures footnotes included. The footnotes were included in the main manuscript while the figures were submitted as separate files, as stated in the journal’s author guidelines.

Reviewer #3:

2. Methodology

• The isolation sources and strain identifiers of the fungal isolates should be more explicitly described to ensure reproducibility.

Response: The endophytic fungal isolates used in the present study were derived from previous studies, and these isolates have been evaluated morphologically and molecularly. This has been mentioned in the revised manuscript. Information on the endophytic fungal isolates has been provided in the revised supplementary information file.

• The parameters of the Nesslerization assay (calibration curve, regression equation, and R² value) should be presented.

Response: All parameters of the Nesslerization assay (calibration curve, regression equation, and R² value) were presented in the supplementary information file. However, they have all been moved to the main manuscript in the revised version.

• Statistical methods (ANOVA, Duncan’s test, t-test) are appropriate but not fully reported. Please specify the sample size (n), number of replicates, degrees of freedom, and exact p-values.

Response: The sample size (number of examined fungal isolates) and the number of replicates were presented in the "Materials and Methods" and "Results" sections, and the degrees of freedom were provided in the analysis of variance tables in the supplementary file. However, the exact p-values have been calculated, and the majority of the analysis of variance tables have been moved to the main manuscript in the revised version. In addition, the sample size and number of replicates have been included in the table notes again. Other information, tables, and graphs are provided in the supplementary file.

• Include a short statement on how data normality and variance homogeneity were verified before using parametric tests.

Response: Data normality and variance homogeneity were verified using the Shapiro-Wilk and Levene’s tests, respectively. This has been mentioned in the revised manuscript.

• The manuscript claims that “all relevant data are within the manuscript and its supporting information,” but the raw data (enzyme activity values, absorbance readings, and statistical outputs) are not included. To comply with the PLOS Data Policy, the authors should upload all underlying datasets as supplementary files or in a public repository.

Response: All statistical outputs, including analysis of variance tables (degrees of freedom and exact p-values) and additional graphs and tables, have been provided in the supplementary information file. The sample size and number of replicates have been stated for each experimental section. The raw data (zone indices of enzyme production, absorbance readings of enzyme activity, absorbance readings at various ammonium concentrations for calibration curve construction, absorbance readings of enzyme activity in culture media without substrate, and zone indices of 10 different culture media) are attached as an Excel file.

• Figures are informative but should include error bars, axis labels, and units.

Response: All figures and graphs have been checked. All of them include error bars, axis labels, and units. Due to the zone index equation, which is calculated as the ratio of the diameter of the color zone to that of the fungal colony, the graphs presenting the zone index of the studied fungal isolates do not have units.

• A summary table or graphical heatmap of the best-performing isolates would improve data clarity.

Response: A heatmap of eleven endophytic fungal isolates, which showed higher L-asparaginase zone indices according to Duncan's post hoc test, with Alternaria brassicae C selected among them, has been constructed and added to the revised manuscript.

• Correlation results should be visualized with a scatterplot.

Response: The scatterplot of the Pearson correlation results has been constructed and added to the revised manuscript.

• Some numerical results (e.g., enzyme activity values) are repeated excessively; consider simplifying.

Response: It has been considered in the revised version.

• The discussion effectively links results to previous studies but occasionally becomes descriptive rather than analytical.

Response: It has been considered in the revised version.

• The statement about glucose concentration inhibiting enzyme activity is speculative; it should be supported by data or moved to “future work.”

Response: The potential inhibition of enzyme production by glucose concentration was only a postulation, and no experiment was conducted. Therefore, it has been removed in the revised manuscript.

• The claim that Alternaria brassicae C is a promising industrial strain is reasonable but needs further biochemical or kinetic validation.

Response: This study was performed to screen some endophytic fungi for their potential to produce L-asparaginase with lower L-glutaminase and urease activities. According to the findings of the present study, Alternaria brassicae C is proposed as a promising isolate among the studied isolates. Naturally, additional in vitro and in vivo validation is necessary for industrial application.

• The manuscript is written in standard, intelligible English, though minor stylistic corrections (sentence length, punctuation, and consistency of scientific names) are recommended. Please italicize all genus and species names and ensure consistent use of units (e.g., U mL⁻¹).

Response: It has been considered in the revised version.

Reviewer #4:

• There are obvious conclusion made which cannot be validated based on the current results. For example, the author claims that the best isolate has low urease activity which is almost 28% of the total activity.

• They concluded that Alternaria L-asparaginase is best one and can be used for commercial production but failed to show for what commercial purpose.

• Eventually the ultimate final end goal is missing. Merely reporting isolation of L-asparaginase production from endophytic fungi cannot solve the purpose.

• There is no comparision of production parameters, size of the enzyme (need to go beyond qualtitative tests). In my opinion, the work is too premature to be published at this point.

Response:

All of the comments were considered in the revised manuscript. The purpose of the present study was to identify a promising endophytic fungal isolate as a eukaryotic candidate for L-asparaginase production with lower L-glutaminase and urease co-activities as side effects. The findings of this study are intended as a suggestion of a new source for L-asparaginase production, which can be considered for further studies to identify an eligible endophytic fungal isolate for pharmaceutical or food industries application in the future. Undoubtedly, in line with and as a continuation of the present study, further in vitro and in vivo studies are necessary, which cannot be included in a single paper.

The goal of the present study has been stated in the main manuscript. Certainly, the purpose of this study will be mentioned more clearly in the revised manuscript.

The evaluation of L-glutaminase and urease in the 11 endophytic fungal isolates with the highest L-asparaginase zone indices, according to Duncan's post hoc test, showed that two isolates (Alternaria brassicae C and Aureobasidium pullulans IH1-2) did not produce L-glutaminase, and A. brassicae C produced urease at a lower level than Au. pullulans IH1-2 (Figs. 1 and 2). For these reasons, among the 62 studied isolates, A. brassicae C has been suggested as a promising endophytic fungal isolate for further studies in the future.

---

## [Decision Letter · Decision Letter 1]

14 Dec 2025

L-asparaginase activity in some endophytic fungi: glutaminase-free and low urease co-activities

PONE-D-25-54924R1

Dear Dr. Fotouhifar,

We’re pleased to inform you that your manuscript has been judged scientifically suitable for publication and will be formally accepted for publication once it meets all outstanding technical requirements.

Kind regards,

Shashi Kant Bhatia

Academic Editor

PLOS One

---

## [Editor Report · Acceptance letter]

PONE-D-25-54924R1

PLOS One

Dear Dr. Fotouhifar,

I'm pleased to inform you that your manuscript has been deemed suitable for publication in PLOS One. Congratulations! Your manuscript is now being handed over to our production team.

Kind regards,

on behalf of

Dr. Shashi Kant Bhatia

Academic Editor

PLOS One